# Photoelectron Yield Spectroscopy and Transient Photocurrent Analysis for Triphenylamine-Based Photorefractive Polymer Composites

**Naoto Tsutsumi** [1,*] , **Yusuke Mizuno** [2] , **Boaz Jessie Jackin** [3] , **Kenji Kinashi** [1] , **Takafumi Sassa** [4] , **Ha Ngoc Giang** [5] **and Wataru Sakai** [1]

1    Faculty of Materials Science & Engineering, Kyoto Institute of Technology, Sakyo, Kyoto 606-8585, Japan
2    Master's Program of Innovative Materials, Graduate School of Science and Technology, Kyoto Institute of Technology, Sakyo, Kyoto 606-8585, Japan
3    Materials Innovation Laboratory, Kyoto Institute of Technology, Sakyo, Kyoto 606-8585, Japan
4    Photonics Control Technology Team, RIKEN Center for Advanced Photonics, Wako 351-0198, Japan
5    Faculty of Chemical Technology, Ho Chi Minh City University of Food Industry, Ho Chi Minh City 72000, Vietnam
*    Correspondence: tsutsumi@kit.ac.jp; Tel.: +81-75-724-7810

**Abstract:** The photocurrent for poly(4-(dimethylamino)benzyl acrylate) (PDAA) photorefractive composites with (4-(diphenylamino)phenyl)methanol (TPAOH) photoconductive plasticizers was measured to be two orders of magnitude higher than that obtained with (2,4,6-trimethylphenyl) diphenylamine (TAA) photoconductive plasticizers. In this study, to determine the reason for the large difference in the photocurrent measured for PDAA photorefractive composites containing two different photoconductive plasticizers of TPAOH and TAA, the highest occupied molecular orbital (HOMO) level identical to the ionization potential ($I_P$) and the width of the density of states (DOS) were evaluated using photoelectron yield spectroscopy, and the transient photocurrent was analyzed using a two-trap model. The estimated hole mobility was also rationalized using a Bässler formalism together with the energetic disorder of the width of the DOS and the positional disorder of the scattering situation for carrier hopping.

**Keywords:** photorefractivity; transient photocurrent; photoelectron yield spectroscopy; width of the density of states; quantum efficiency for photocarrier generation; hole mobility; trap parameters

## 1. Introduction

Since the discovery of photorefractive (PR) polymers in 1991 [1], these materials have attracted attention in the field of organic and polymer optoelectronics, including nonlinear optics and organic photonic materials, and during the past two decades, over 700 research papers for PR polymers have been published per year [2].

The PR properties of polymers are based on space-charge formation due to photoconductive and first-order optoelectronic effects (Pockels effect) in PR polymers [3]. In general, PR polymers consist of a photoconductive polymer, a nonlinear optical (NLO) chromophore, a photoconductive or inert plasticizer, and a sensitizer [4]. The photoconductive polymer and sensitizer cooperatively work to produce mobile positive charge carriers (holes) and immobilized negative charges (electrons) following light illumination. Holes are transported through the photoconductive manifold and trapped. Immobilized electrons are localized at the sensitizer as anions or anion radicals. The separated holes and electrons form an internal space-charge field, which is the essence of the physical quantity of the PR effect. With an externally applied field, the internally formed space-charge field leads to nonlinear refractive index modulation via both the Pockels effect and the molecular orientation of the NLO chromophore along the interference illumination pattern. The

noncentrosymmetric alignment and molecular orientation of the NLO chromophore are key for the PR response.

Furthermore, the photoconductive properties of PR polymers are important for investigating and understanding the trapping event for hole carriers and, thus, space-charge field formation [5,6]. We also investigated the correlation between the photorefractive response and photoconductivity for poly[bis(2,4,6-trimethylphenyl)amine] (PTAA)-based PR polymers [7–11]. Based on the large hole mobility of the order of $10^{-3}$ to $10^{-2}$ cm$^2$ V$^{-1}$ s$^{-1}$ for PTAA due to high hole mobility, the PTAA-based PR polymer has a response time of the order of hundreds of microseconds with a high optical diffraction efficiency of over 50% and a low trap density of the order of $10^{14}$ cm$^{-3}$ and, thus, a very low space-charge field of less than 1 V $\mu$m$^{-1}$. Based on the theory of the formation and diminishing of space-charge gratings in photoconductive polymers [12], a two-trap model with shallow and deep traps has been developed [13]. The space-charge dynamics for poly(N-vinylcarbazole) (PVK)-based PR polymers [13] and the photocurrent dynamics for poly(phenylene vinylene)-based PR polymers have also been investigated using a two-trap model [14]. Quantitative analysis of the transient photocurrent has been carried out for poly(4-diphenylamino) styrene (PDAS)-based PR polymers using a two-trap model [15]. Transient photocurrents were reported for poly(4-(diphenylamino)benzyl acrylate) (PDAA)-based PR polymers [16,17]. In particular, a significant difference in the photocurrent of two orders of magnitude has been reported for PDAA PR composites with different photoconductive plasticizers, (4-(diphenylamino)phenyl)methanol (TPAOH) and (2,4,6-trimethylphenyl)diphenylamine (TAA) [17]. However, a detailed analysis of the photocurrent in PDAA PR composites has not yet been performed.

In this report, we have used photoelectron yield spectroscopy (PYS) and analysis of the transient photocurrent using a two-trap model [13,14] to clarify the significant difference in photocurrents for PDAA PR composites with TPAOH and TAA photoconductive plasticizers. PYS is a useful tool to evaluate the ionization potential of molecules and composites. The detailed analysis of transient photocurrents also provides useful insight into the trapping events in the PR composite. The photophysicochemical roles of both TPAOH and TAA in the photocarrier generation process and the roles of TPAOH and TAA in hole transport and trapping behavior are investigated and discussed.

## 2. Materials and Methods

PDAA as a hole transport polymer, TPAOH or TAA as a photoconductive plasticizer, (4-(azepan-1-yl)-benzylidene)malononitrile (7-DCST) as a NLO dye, and [6,6]-phenyl-C61-butyric acid methyl ester (PCBM) as an electron acceptor were used. Mixture of PDAA, TPAOH or TAA, 7-DCST, and PCBM was dissolved in tetrahydrofuran (THF), and then the THF solution was cast on a hot plate to prepare PR composite. Obtained PR composite was pressed between indium-tin-oxide (ITO)-coated glass plates to prepare the PR composite sample film. The structural formulas for the compounds and the details for the preparation of the PDAA-based polymers are shown in previous papers [17]. The thickness of the sample film was 50 $\mu$m.

Photoelectron yield spectroscopy (PYS) was monitored every 0.02 eV in the range of 4.0 eV and 9.5 eV in vacuum using a Bunkokeiki BIP-KV202GTGK PYS instrument. The light source is deuterium lamp (D2 lamp). The density of states (DOS) was determined by the first derivative of PYS data. Composite material (5 mg) was dissolved in 0.2 mL mixture solvent of toluene and cyclohexane (4/1, vol%). Sample film was spin-coated onto an ITO-coated glass plate from the solution at 1000 rpm for 60 s. After spin-coating, the sample film was dried at 52 °C for 13 h. Three times measurements were performed at different illumination positions for the same sample.

Transient photocurrent was measured using a picoammeter (6485, Keithly, Solon, OH, USA) and data acquisition by a Lecroy 6051A digital oscilloscope under illumination of 640 nm laser with 400 mWcm$^{-2}$ at an applied electric field of 40 V$\mu$m$^{-1}$. Laser source is iFLEX2000, QIOPTIQ. The photocurrent signal was monitored under illumination at

1 s followed by under unillumination at 1 s. Then, repeated measurements of under illumination at 1 s and under unillumination at 1 s were performed 4 times. Total repeated measurements were 5 times.

## 3. Results and Discussion

### 3.1. Photoelectron Yield Spectroscopy and Energy Diagram

In a previous report [17], we measured a photocurrent of 4.2 μA for PDAA/TPAOH/ 7-DCST/PCBM and 0.036 μA for PDAA/TAA/7-DCST/PCBM at E = 40 V μm$^{-1}$. To determine the significant difference in the photocurrent between PDAA/TPAOH/7-DCST/ PCBM and PDAA/TAA/7-DCST/PCBM, the role of TPAOH and TAA should be clarified in photorefractive composites. The molecular structures of TPAOH and TAA are shown in Figure 1. The main framework of the molecular structure of TPAOH is almost the same as that of the PDAA monomer. However, TAA has three bulky methyl moieties attached to one phenyl group. Ionization potentials for donors of PDAA, TPAOH, and TAA of 5.69 eV, 5.64 eV, and 5.90 eV, respectively, have been reported previously [8,16]. These ionization potentials ($I_p$) correspond to the highest occupied molecular orbital (HOMO) level, and, thus, the negative numeral of $I_p$ is a HOMO level for these donors. However, these numerals do not directly tell us how the ionization state (HOMO level state) is formed in the composites. Thus, we need to know the difference in the ionization state (HOMO level state) between PDAA in the presence of TPAOH and in the presence of TAA.

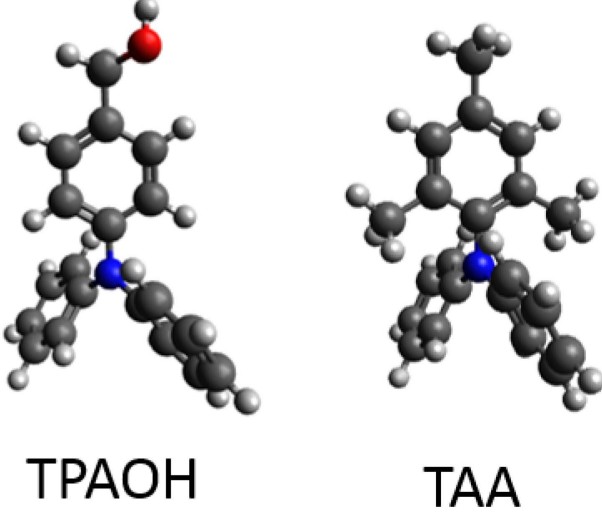

**Figure 1.** Molecular structures of TPAOH and TAA energetically stabilized using a molecular orbital simulation. Theoretical calculations with the Guassian09 package software, using the functional/basis set RB3LYP, were applied.

Photoelectron yield spectroscopy is a useful tool for evaluating the ionization potentials (HOMO levels) and the width of the density of states (DOS). The photoelectron yield is plotted as a function of photon energy for PDAA/TPAOH (50/50) and PDAA/TAA (50/50) in Figure 2. The inserted figures show plots of the photoelectron yield over the entire measured photon energy range from 4.0 to 9.5 eV. The photoelectron yield$^{1/3}$ linearly increases above the threshold. The ionization potential and the HOMO level are determined from the threshold at which the photoelectron yield$^{1/3}$ linearly increases from the baseline. The photoelectron yield$^{1/3}$ is increased by 5.78 eV (averaged) for PDAA/TPAOH and 5.79 eV (averaged) for PDAA/TAA with increasing photon energy. These numerals are identified with the ionization potential (HOMO level) of the PDAA in the presence of TPAOH and in the presence of TAA. The ionization potential of PDAA/TPAOH is close to that of PDAA/TAA.

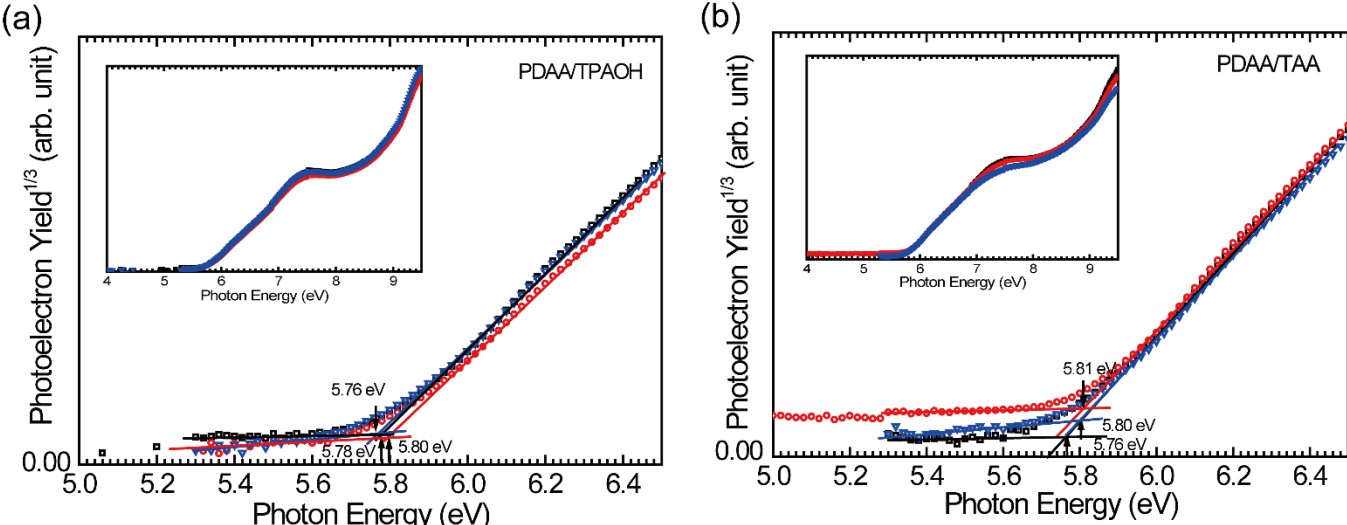

**Figure 2.** Plots of photoelectron yield as a function of incidence photon energy for (**a**) PDAA/TPAOH and (**b**) PDAA/TAA. Three times measurements were performed at different illumination positions for each sample. Each measurement is separately shown by black, red, and blue plots. Black arrows and numerals indicate the ionization potential for each sample.

The photoelectron yield is plotted as a function of photon energy for PDAA/TPAOH/ 7-DCST/PCBM (35/35/30/0.6) and PDAA/TAA/7-DCST/PCBM (35/35/30/0.6) in Figure 3. The photoelectron yield$^{1/3}$ is increased by 5.80 eV (averaged) for PDAA/TPAOH/ 7-DCST/PCBM and by 5.72 eV (averaged) for PDAA/TAA/7-DCST/PCBM with increasing photon energy. Thus, the ionization potential (HOMO level) of PDAA/TPAOH/ 7-DCST/PCBM is 5.80 eV (−5.80 eV), and that of PDAA/TAA/7-DCST/PCBM is 5.72 eV (−5.72 eV). It is noted that the HOMO level of −5.72 eV for PDAA/TAA/7-DCST/PCBM is higher than that of −5.80 eV for PDAA/TPAOH/7-DCST/PCBM.

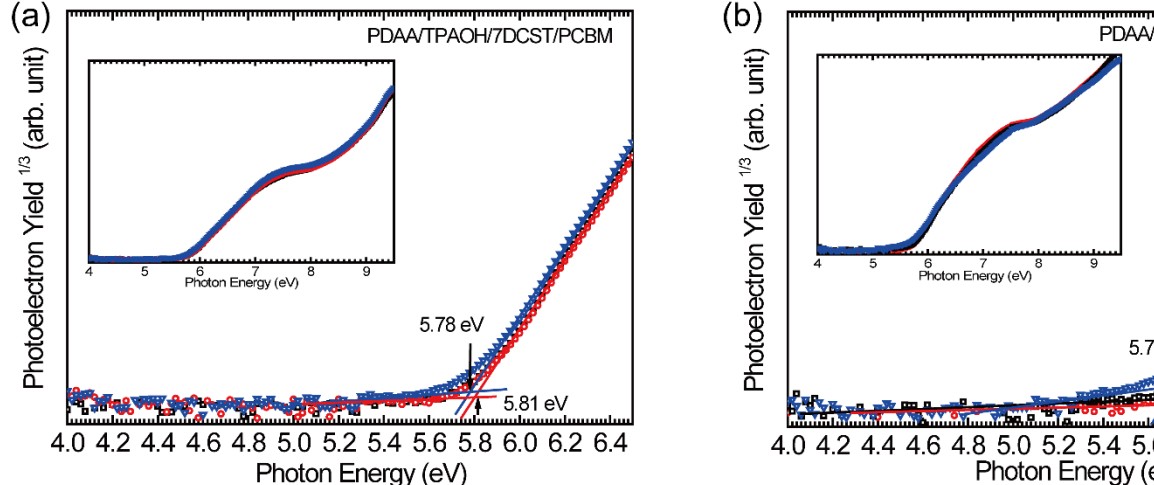

**Figure 3.** Plots of photoelectron yield as a function of incidence photon energy (**a**) for PDAA/TPAOH/7-DCST/PCBM and (**b**) for PDAA/TAA/7-DCST/PCBM. Three times measurements were performed at different illumination positions for each sample. Each measurement is separately shown by black, red, and blue plots. Black arrows and numerals indicate the ionization potential for each sample.

The energy diagram for both systems is illustrated in Figure 4. As shown in Figure 4, the HOMO level for PDAA/TAA/7-DCST/PCBM is −5.72 eV, which is a considerably higher HOMO level even though each component except for PDAA has a lower HOMO

level. In contrast, the HOMO level for PDAA/TPAOH/7-DCST/PCBM is −5.80 eV, which is close to the average value of the HOMO level of each component.

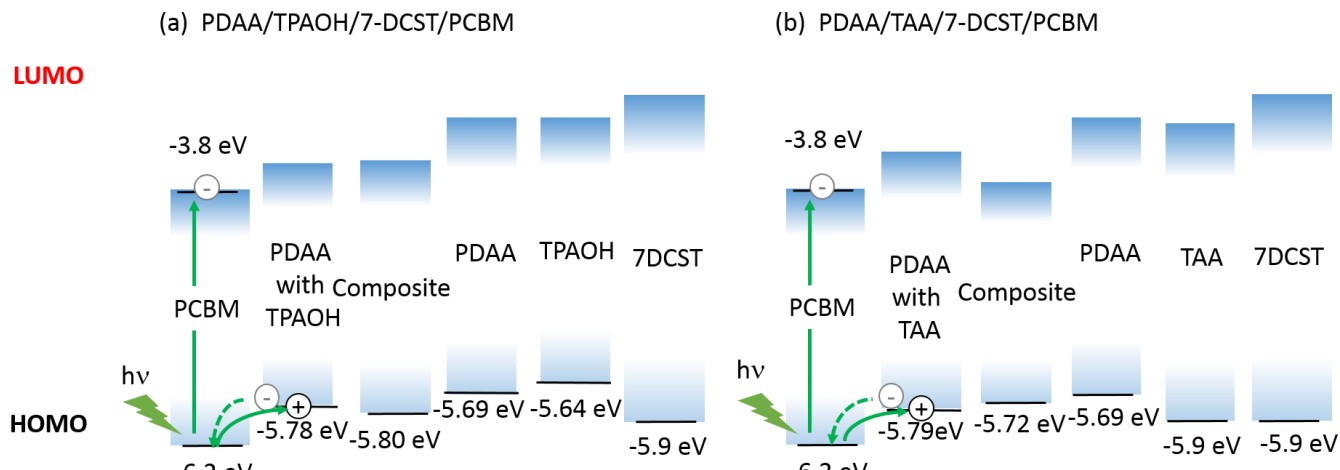

**Figure 4.** Energy diagram (HOMO and LUMO level) for (**a**) PDAA/TPAOH/7-DCST/PCBM and (**b**) PDAA/TAA/7-DCST/PCBM.

### 3.2. Evaluation of the Transient Photocurrent Determined Using Two-Trap Model

We analyzed the transient photocurrents for PDAA/TPAOH/7-DCST/PCBM and PDAA/TAA/7-DCST/PCBM using a two-trap (shallow and deep traps) model [13,14]. To explain the photorefractive dynamics, a one-trap (shallow trap) model [12] is proposed, and a modified two-trap (shallow and deep traps) model [13,14] is proposed. The modified model satisfies the following nonlinear equations for the photorefractive dynamics:

$$J_{\text{ph}} = e\mu\rho E - eD\frac{\partial\rho}{\partial x} \tag{1}$$

$$\frac{\partial\rho}{\partial t} = \frac{\partial N_{\text{A}}^-}{\partial t} - \frac{\partial T^+}{\partial t} - \frac{\partial M^+}{\partial t} - \frac{1}{e}\frac{\partial J_{\text{ph}}}{\partial x} \tag{2}$$

$$\frac{\partial E}{\partial x} = \frac{e}{\varepsilon_0\varepsilon_{\text{r}}}\left(\rho + T^+ + M^+ - N_{\text{A}}^-\right) \tag{3}$$

$$\frac{\partial T^+}{\partial t} = \gamma_{\text{T}}\left(T - T^+\right)\rho - \beta_{\text{T}}T^+ \tag{4}$$

$$\frac{\partial M^+}{\partial t} = \gamma_{\text{M}}\left(M - M^+\right)\rho - \beta_{\text{M}}M^+ \tag{5}$$

$$\frac{\partial N_{\text{A}}^-}{\partial t} = sI\left(N_{\text{A}} - N_{\text{A}}^-\right) - \gamma_{\text{R}}N_{\text{A}}^-\rho \tag{6}$$

where $J_{\text{ph}}$ is the current density; $e$ is the elementary charge; $\mu$ is the mobility of the charge carriers; $\rho$ is the charge carrier density; $E$ is the electric field; $D$ is the diffusion coefficient; $\varepsilon_0$ is the dielectric permittivity in space; $\varepsilon_{\text{r}}$ is the dielectric constant; $N_{\text{A}}$, $T$, and $M$ are the total density of sensitizers, shallow traps, and deep traps, respectively; $N_{\text{A}}^-$, $T^+$, and $M^+$ are the density of sensitizer anions, filled shallow traps, and filled deep traps, respectively; $s$ is the photogeneration cross-section; $\gamma_{\text{T}}$ is the shallow trapping rate; $\gamma_{\text{M}}$ is the deep trapping rate; $\gamma_{\text{R}}$ is the recombination rate; $\beta_{\text{T}}$ is the detrapping rate from the shallow traps; $\beta_{\text{M}}$ is the detrapping rate from the deep traps; and $I$ is the intensity of the light illumination. The photogeneration cross-section $s$ is given by $s = \phi\alpha\lambda/(hcN_{\text{A}})$, where $\phi$ is the quantum efficiency (*QE*) for photocarrier generation, $\alpha$ is the absorption coefficient, $\lambda$ is the wavelength of the light, $h$ is the Planck constant, and $c$ is the speed of light.

For hole transport, the diffusion of the hole is negligibly small compared with the electric-field-dependent drift mobility in the amorphous polymer matrix. In the previous

study [13,14], the *D* parameter was also neglected. Thus, in the simulation process to reproduce the measured transient photocurrent, we need to select reasonable parameters: the quantum efficiency for photocarrier generation *QE*, hole mobility $\mu$, the shallow trapping rate $\gamma_T$, the density of shallow trap *T*, the detrapping rate from the shallow traps $\beta_T$, the deep trapping rate $\gamma_M$, the density of deep trap *M*, the detrapping rate from the deep traps $\beta_M$, and the recombination coefficient $\gamma_R$. Even though the trapping parameters for the shallow and deep traps are almost fixed, a wide range of *QE*, hole mobility $\mu$, and recombination coefficient $\gamma_R$ can be reproduced in the measured transient photocurrent as listed in Table A1 in Appendix A. Thus, *QE* or hole mobility $\mu$ should be first determined by other data. The recombination coefficient $\gamma_R$ is proportional to hole mobility with the Langevin recombination process.

In the present simulation, the quantum efficiency for photocarrier generation *QE*, the density of shallow trap *T*, and the density of deep trap *M* were determined from the photorefractive data. The trap density of $1.2 \times 10^{16}$ cm$^{-3}$ for the shallow trap and $2.0 \times 10^{16}$ cm$^{-3}$ for the deep trap for PDAA/TPAOH/7-DCST/PCBM, and $1.1 \times 10^{16}$ cm$^{-3}$ for the shallow trap for PDAA/TAA/7-DCST/PCBM are reasonably evaluated. The total density of traps is reasonably comparable to the photorefractive number density of traps determined using the Kukhtarev model [18], $1.4$–$3.1 \times 10^{16}$ cm$^{-3}$, reported previously for PDAA PR composites [16]. Therefore, *QE* is determined, and a reasonable trap density is used; we can reproduce the measured photocurrent with proper hole mobility. *QE* is determined from the photorefractive response time as follows. The difference in the photorefractive performance of PDAA/TAA/7-DCST/PCBM and PDAA/TPAOH/7-DCST/PCBM in terms of the optical diffraction and the response time are discussed from the aspect of trapping behavior. The photorefractive quantities of diffraction efficiency and response time reported previously [17] are summarized in Table 1. The optical diffraction efficiency for both polymers is comparable, but the response time $\tau$ of 8 ms for PDAA/TPAOH/7-DCST/PCBM is faster than that of 67 ms for PDAA/TAA/7-DCST/PCBM.

**Table 1.** Summary of the photorefractive parameters of diffraction efficiency, response time, the absorption coefficient, and the evaluated *QE* for different plasticizers.

| Sample | $\eta$ (%) | $\tau$ (ms) | $\alpha_{532}/\alpha_{640}$ | *QE* |
|---|---|---|---|---|
| PDAA/TPAOH/7-DCST/PCBM (35/35/30/0.6) | $39 \pm 1$ | $8 \pm 0.8$ | 200/59 | $4.3 \times 10^{-3}$ |
| PDAA/TAA/7-DCST/PCBM (35/35/30/0.6) | $75 \pm 0.8$ | $67 \pm 0.6$ | 134/45 | $7.0 \times 10^{-4}$ |

$E = 45$ V $\mu$m$^{-1}$; wavelength, 532 nm; laser power, 650 mW cm$^{-2}$ [16].

The response time (growth time) $\tau$ is defined as the time needed to fill the trap by the photogenerated holes [6]:

$$\tau = \frac{T_i}{\left(\frac{\alpha \phi I \lambda}{hc}\right)} \tag{7}$$

where $\tau$ is the response time, and $T_i$ is the initial trap density in Schildkraut's trapping model [12]. Thus, we can estimate the quantum efficiency for the photogeneration of charge carriers $\phi$ using Equation (7) with the observed response time and the trap density. The *QE* for carrier photogeneration estimated using Equation (7) with the density of the shallow traps and the response time is given in Table 1. The *QE* for PDAA/TPAOH/7-DCST/PCBM and PDAA/TAA/7-DCST/PCBM is determined to be $4.3 \times 10^{-3}$ and $7.0 \times 10^{-4}$, respectively.

To reproduce the shape of the transient photocurrents, we first focus on the shoulder or the plateau of the transient current i− wide time range from 0.001 s to 0.1 s. PDAA/TPAOH/7-DCST/PCBM has a very narrow shoulder, around 0.001 s, as shown in Figure 5a, whereas PDAA/TAA/7-DCST/PCBM has a wide plateau of the transient photocurrent from 0.001 to 0.1 s (or 10 s), as shown in Figure 6a. The second focusing point is whether the significant change in the transient photocurrent above 0.1 s occurs or not. The transient photocurrent for PDAA/TPAOH/7-DCST/PCBM significantly decreases above

0.1 s, as shown in Figure 5a, whereas that of PDAA/TAA/7-DCST/PCBM is almost flat above 0.1 s, as shown in Figure 6a. Then, the transient photocurrent for PDAA/TPAOH/7-DCST/PCBM is governed by a shallow trapping event followed by a deep trapping event, whereas PDAA/TAA/7-DCST/PCBM by a shallow trapping event (less contribution of a deep trapping event). Then, with the above idea, we think that the key parameters are the trapping rate and the detrapping rate from the shallow and deep traps to reproduce the transient photocurrent for both cases. For PDAA/TPAOH/7-DCST/PCBM, the detrapping rate from the shallow traps $\beta_T$ should be a little lower or comparable to the trapping rate of the shallow trap (the product of $\gamma_T$ and $T$), and the detrapping rate from the deep trap $\beta_M$ should be much lower than the trapping rate of the deep trap (the product of $\gamma_M$ and $M$). Conversely, the transient photocurrent for PDAA/TAA/7-DCST/PCBM is solely governed by the shallow trapping event, and the deep trapping effect should be negligibly small.

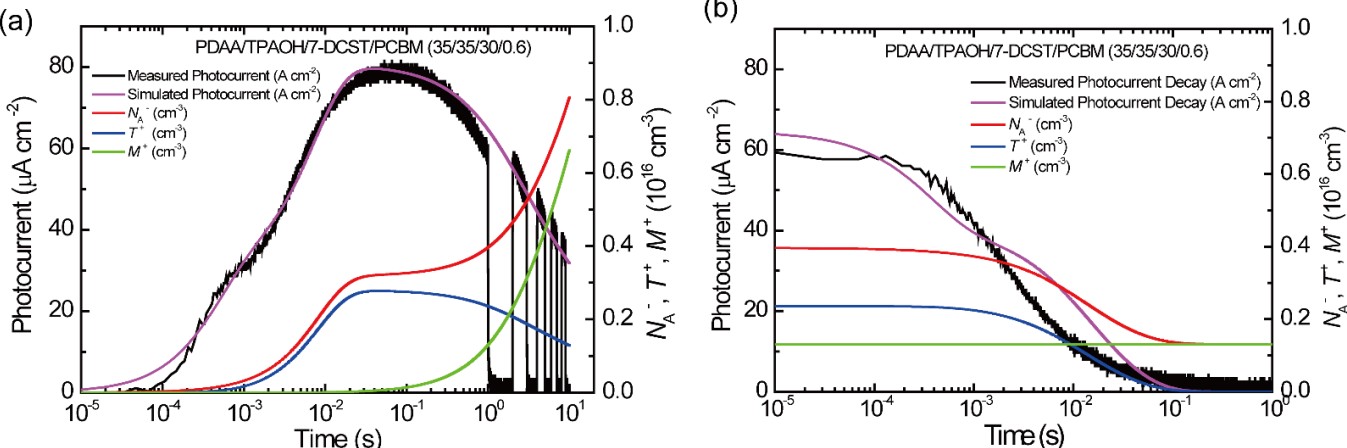

**Figure 5.** (**a**) Rising transient photocurrent (black plots) and the simulated photocurrent (pale purple curve) for PDAA/TPAOH/7-DCST/PCBM: red curve, transient density for the sensitizer anion $N_A^-$; blue curve, transient density for filled shallow traps $T^+$; green curve, transient density for filled deep traps $M^+$; (**b**) decay profile.

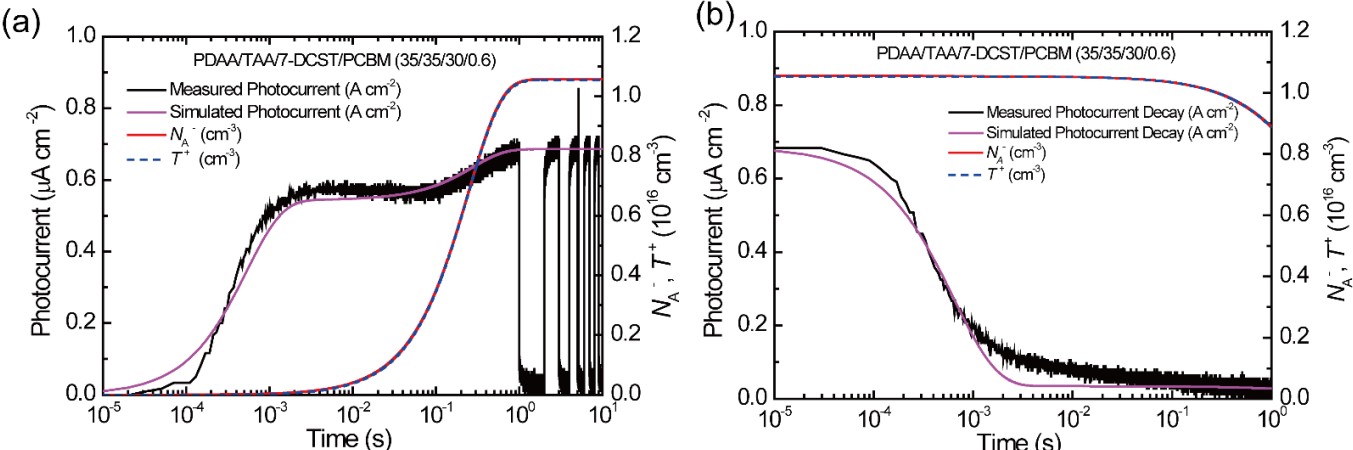

**Figure 6.** (**a**) Rising transient photocurrent (black plots) and the simulated photocurrent (pale purple curve) for PDAA/TAA/7-DCST/PCBM: red curve, transient density for the sensitizer anion $N_A^-$; blue curve, transient density for filled shallow traps $T^+$; (**b**) decay profile.

Then, the transient photocurrents for PDAA/TPAOH/7-DCST/PCBM (35/30/30/0.6) and PDAA/TAA/7-DCST/PCBM (35/30/30/0.6) are simulated using the parameters listed in Table 2, respectively. The simulated transient density for $N_A^-$, $T^+$, $M^+$, and the photocurrent and decaying photocurrent after blocking illumination for PDAA/TPAOH/

7-DCST/PCBM is plotted as a function of the logarithmic timescale in Figure 5a,b, respectively. The same type of simulation is shown in Figure 6a,b for PDAA/TAA/7-DCST/PCBM.

**Table 2.** Summary of the simulated parameters of quantum efficiency (*QE*) for photocarrier generation, hole mobility, trapping, and recombination parameters for PDAA/TPAOH/7-DCST/PCBM (35/35/30/0.6) and PDAA/TAA/7-DCST/PCBM (35/35/30/0.6).

| | | | PDAA/TPAOH/7-DCST/PCBM (35/35/30/0.6) | | | | | | |
|---|---|---|---|---|---|---|---|---|---|
| $QE/\alpha$ (cm$^{-1}$) | $\mu$ (cm$^2$ V$^{-1}$ s$^{-1}$) | $\gamma_T$ (cm$^3$ s$^{-1}$) | $T$ (cm$^{-3}$) | $\beta_T$ (s$^{-1}$) | $\gamma_M$ (cm$^3$ s$^{-1}$) | $M$ (cm$^{-3}$) | $\beta_M$ (s$^{-1}$) | $\gamma_R$ (cm$^3$ s$^{-1}$) |
| $4.3 \times 10^{-3}/59$ | $3.3 \times 10^{-6}$ | $1.6 \times 10^{-13}$ | $1.2 \times 10^{16}$ | 200 | $2.0 \times 10^{-16}$ | $2.0 \times 10^{16}$ | 0.01 | $2.7 \times 10^{-13}$ |
| | | | PDAA/TAA/7-DCST/PCBM (35/35/30/0.6) | | | | | | |
| $QE/\alpha$ (cm$^{-1}$) | $\mu$ (cm$^2$ V$^{-1}$ s$^{-1}$) | $\gamma_T$ (cm$^3$ s$^{-1}$) | $T$ (cm$^{-3}$) | $\beta_T$ (s$^{-1}$) | $\gamma_M$ (cm$^3$ s$^{-1}$) | $M$ (cm$^{-3}$) | $\beta_M$ (s$^{-1}$) | $\gamma_R$ (cm$^3$ s$^{-1}$) |
| $7.0 \times 10^{-4}/45$ | $3.9 \times 10^{-7}$ | $1.7 \times 10^{-13}$ | $1.1 \times 10^{16}$ | 0.2 | - | - | - | $1.4 \times 10^{-13}$ |

Measurement conditions: $N_A$ = (PCBM) = $4.76 \times 10^{18}$ cm$^{-3}$; $E$ = 40 V μm$^{-1}$; wavelength, 640 nm; laser power, 400 mW cm$^{-2}$.

### 3.3. Analysis of the Trapping Behavior and Transient Density for Filled Traps

Both systems show almost the same values for the total density of the shallow traps $T$ of $1.1$–$1.2 \times 10^{16}$ cm$^{-3}$, a shallow trapping rate $\gamma_T$ of $1.6$–$1.7 \times 10^{-13}$ cm$^3$ s$^{-1}$, and a recombination coefficient $\gamma_R$ of $1.4$–$2.7 \times 10^{-13}$ cm$^3$ s$^{-1}$. The total density of the deep traps $M$ is $2.0 \times 10^{16}$ cm$^{-3}$ for PDAA/TPAOH/7-DCST/PCBM. The total density of the traps is comparable to the photorefractive number density of traps determined using the Kukhtarev model [18], $1.4$–$3.1 \times 10^{16}$ cm$^{-3}$, reported previously for PDAA PR composites [16].

For the shallow trapping event, the trapping rate of the shallow trap (the product of $\gamma_T$ and $T$) for PDAA/TPAOH/7-DCST/PCBM of 1920 s$^{-1}$ is comparable with that obtained for PDAA/TAA/7-DCST/PCBM, 1870 s$^{-1}$. On the other hand, the detrapping rate $\beta_T$ for PDAA/TPAOH/7-DCST/PCBM of 200 s$^{-1}$ is much faster than that obtained for PDAA/TAA/7-DCST/PCBM, 0.2 s$^{-1}$. A faster tapping rate of 1870 s$^{-1}$ and a slower detrapping rate $\beta_T$ of 0.2 s$^{-1}$ contributed to the almost flat and plateau photocurrent measured in the time range of 0.001 to 10 s for PDAA/TAA/7-DCST/PCBM.

For the deep trapping event, the trapping rate of the deep trap (the product of $\gamma_M$ and $M$) of 4.0 s$^{-1}$ and the detrapping rate $\beta_M$ from the deep trap of 0.01 s$^{-1}$ significantly contributed to the large decrease in the transient photocurrent in the time range beyond 0.1 s for PDAA/TPAOH/7-DCST/PCBM. On the other hand, the contribution of the deep trapping event to the transient photocurrent is negligibly small for PDAA/TAA/7-DCST/PCBM. In other words, the transient photocurrent for PDAA/TAA/7-DCST/PCBM can be described by the one-trap model.

Here, the transient densities of the filled shallow and deep traps shown in Figures 5a and 6a are compared for both systems. For PDAA/TPAOH/7-DCST/PCBM, the initial increase in the photocurrent is mainly governed by the transient density of the filled shallow traps (blue curve in Figure 5a) at 0.1 s, but beyond 0.1 s, the role of the transient density of the filled shallow traps decreases and that for filled deep traps (green curve in Figure 5a) increases. On the other hand, the entire photocurrent for PDAA/TAA/7-DCST/PCBM is mainly governed by the transient density of the filled shallow traps, and the contribution of the filled deep traps is negligibly small. These results suggest that the initial transient photocurrent for PDAA/TPAOH/7-DCST/PCBM is mainly governed by the detrapping behavior from the shallow traps, which is followed by that from the deep trap at a later time, whereas that for PDAA/TAA/7-DCST/PCBM is governed only by the detrapping behavior from the shallow trap.

The contribution to the transient photocurrent decay is the detrapping of filled traps from the shallow trap in the shorter time region in the time range from $10^{-5}$ to $10^{-1}$ s, which is followed by the detrapping of filled traps from the deep trap in the time range

beyond $10^{-1}$ s for PDAA/TPAOH/7-DCST/PCBM. In contrast, the detrapping of the hole carriers from the shallow trap contributes to the transient photocurrent decay for PDAA/TAA/7-DCST/PCBM.

### 3.4. Relationship between Trapping Behavior and Photorefractive Response

The calculated transient densities for the sensitizer anion, filled shallow traps, and filled deep traps are shown in Figures 5a and 6a. As shown in Figure 5a for PDAA/TPAOH/ 7-DCST/PCBM, the filled shallow trap density (blue curve in the Figure 5a) starts increasing at the time 1 ms, levels out at the time beyond 10 ms, and decreases at the time after 0.1 s followed by the beginning of a large increase in the density of the filled deep traps (green curve shown in the Figure 5a). As shown in Figure 6a, however, the filled shallow trap density starts increasing at the time 10 ms and levels out at the time beyond 1 s for PDAA/TAA/7-DCST/PCBM. The density of the filled shallow traps for PDAA/TPAOH/ 7-DCST/PCBM is $1.70 \times 10^{15}$ cm$^{-3}$ at a response time of 8 ms, whereas the density of the filled shallow traps for PDAA/TAA/7-DCST/PCBM is $2.43 \times 10^{15}$ cm$^{-3}$ at a response time of 67 ms. These results explain that the response time for optical diffraction is given by the time taken to fill a sufficient density of shallow traps to form the space-charge field. In other words, these filled shallow traps work as effective photorefractive traps.

The hole mobility for PDAA/TPAOH/7-DCST/PCBM is determined to be $3.3 \times 10^{-6}$ cm$^2$ V$^{-1}$ s$^{-1}$ with $QE = 4.3 \times 10^{-3}$ and that for PDAA/TAA/7-DCST/PCBM is determined to be $3.9 \times 10^{-7}$ cm$^2$ V$^{-1}$ s$^{-1}$ with $QE = 7.0 \times 10^{-4}$. PDAA/TAA/ 7-DCST/PCBM shows a hole mobility of $3.9 \times 10^{-7}$ that is one order slower than that of $3.3 \times 10^{-6}$ cm$^2$ V$^{-1}$ s$^{-1}$ obtained for PDAA/TPAOH/7-DCST/PCBM.

### 3.5. Estimation of Value for Trap State

We can estimate the value for the trap state (unit is eV) from the detrapping rate $\beta$. The inverse of the detrapping rate $\beta$ is correlated to the time in which the hole carriers are residing in the trap, the trap residing time $t_{\mathrm{tr}}$. The trap residing time is expressed as

$$t_{\mathrm{tr}} = \frac{a}{v} \exp\left(\frac{\Delta E}{kT}\right) = \frac{1}{\beta} \tag{8}$$

where $a$ is the average hopping distance (the average distance between hopping sites), $v$ is the hopping velocity, $\Delta E$ is the value for the trap state, $k$ is the Boltzmann constant, and $T$ is absolute temperature [19]. Hopping velocity is related to drift mobility $\mu$ as

$$\mu = \frac{v}{E} \tag{9}$$

where $E$ is the electric field. With Equations (8) and (9), we can estimate the value for the trap state for both systems. $\Delta E = 0.29$ eV for the shallow trap and $\Delta E = 0.54$ eV for the deep trap were evaluated for PDAA/TPAOH/7-DCST/PCBM. $\Delta E = 0.41$ eV for the shallow trap was evaluated for PDAA/TAA/7-DCST/PCBM. These values for the trap state are reasonable.

### 3.6. DOS Width and Hole Mobility

DOS spectra were estimated by differentiating the measured photoelectron yield spectra as a function of the incident photon energy. DOS curves as a function of the photon energy are shown in Figure 7. The edge part of the DOS curve at a low photon energy region is useful for evaluating the energy dispersion of the carrier hopping sites. To evaluate the width of DOS for the hole hopping sites, the peak separation method was performed using a peak separation analytical tool from Origin 6.1 software. The original DOS curve is presented by a black solid curve, and the separated Gaussian curve is presented by a red dashed curve. The DOS width of the carrier transport manifold for both systems is evaluated from the separated Gaussian peak with the lowest photon energy (red dashed curve shown in Figure 7). The DOS widths for PDAA/TPAOH/7-DCST/PCBM and

PDAA/TAA/7-DCST/PCBM are determined to be 0.138 eV and 0.153 eV, respectively, and are listed in Table 3. The DOS width is significantly related to the energetic disorder, and the energetic disorder can be evaluated from the DOS width. The broader width of DOS is related to a greater energetic disorder and more broadened energetics.

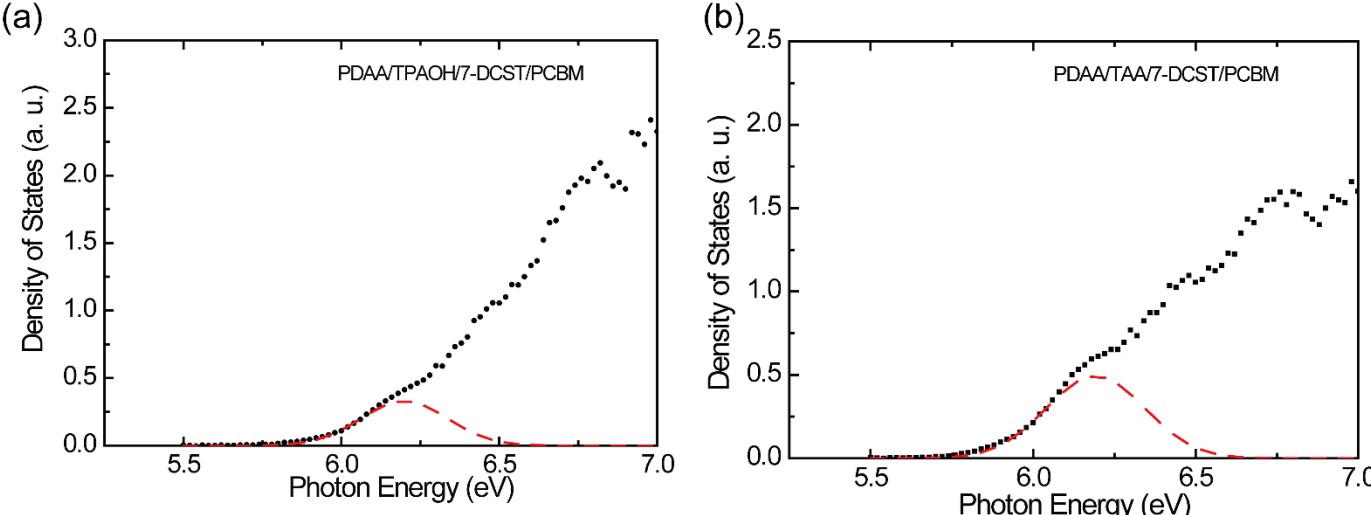

**Figure 7.** DOS curves are plotted as a function of photon energy for (**a**) PDAA/TPAOH/ 7-DCST/PCBM and (**b**) PDAA/TAA/7-DCST/PCBM: black solid curve, measured DOS curve; red curve, separated Gaussian curve at low photon energy region.

**Table 3.** Summary of the DOS width and hole mobility.

| | DOS Width (eV) | $\mu$ [1] (cm$^2$ V$^{-1}$ s$^{-1}$) | $\mu$ [2] (cm$^2$ V$^{-1}$ s$^{-1}$) |
|---|---|---|---|
| PDAA/TPAOH/7DCST/PCBM (35/35/30/0.6) | 0.138 | $3.4 \times 10^{-6}$ | $3.3 \times 10^{-6}$ |
| PDAA/TAA/7DCST/PCBM (35/35/30/0.6) | 0.153 | $4.0 \times 10^{-7}$ | $3.9 \times 10^{-7}$ |

[1] Hole mobility was evaluated using Equation (10). [2] Hole mobility was evaluated from the transient photocurrent.

The carrier hopping rate between adjacent hopping transport molecules (transport manifold) significantly depends on the number density of hopping sites, their ionization potential, and their energetic and spatial (positional) disorder. In Bässler's formalism with a diagonal disorder characterized by a standard deviation of $\sigma$, the variance in the Gaussian energy distribution for the hopping site manifold (energetic disorder) and off-diagonal disorder characterized by the positional disorder described by the parameter $\Sigma$ [20], Monte Carlo simulations result in the following universal law:

$$\mu(E, T) = \mu_0 \exp\left[-\left(\frac{2}{3}\frac{\sigma}{kT}\right)^2\right] \exp\left\{C\left[\left(\frac{\sigma}{kT}\right)^2 - \Sigma^2\right]E^{1/2}\right\} \tag{10}$$

Here, $\sigma$ is the variance in the hopping site energies, $\Sigma$ is a parameter that describes the degree of positional disorder, $\mu_0$ is the prefactor mobility, and $C$ is an empirical constant [20]. Equation (10) is valid for a high electric field on the order of a few tens of V $\mu$m$^{-1}$ and $T_g > T > T_c$, where $T_g$ is the glass transition temperature, and $T_c$ is the nondispersive-to-dispersive transition temperature [21].

The hole mobility is evaluated at $E$ = 40 V $\mu$m$^{-1}$ using Equation (10). For PDAA/ TPAOH/7DCST/PCBM, $\mu$ is evaluated to be $3.4 \times 10^{-6}$ cm$^2$ V$^{-1}$ s$^{-1}$ with a DOS width and parameters of $C = 5.3 \times 10^{-4}$ cm$^{1/2}$V$^{-1/2}$, $\mu_0 = 0.01$ cm$^2$ V$^{-1}$ s$^{-1}$, and $\Sigma = 3.8$; for PDAA/TAA/7DCST/PCBM, $\mu$ is evaluated to be $4.0 \times 10^{-7}$ cm$^2$ V$^{-1}$ s$^{-1}$ with a DOS width and parameters of $C = 5.3 \times 10^{-4}$ cm$^{1/2}$V$^{-1/2}$, $\mu_0 = 0.01$ cm$^2$ V$^{-1}$ s$^{-1}$, and $\Sigma = 4.32$,

as listed in Table 3. These parameters are reasonable for photorefractive polymers [22]. In addition to the higher energetic disorder of the larger width of DOS, the broader positional disorder of the larger $\Sigma$ value is also evaluated for PDAA/TAA/7-DCST/PCBM compared with PDAA/TPAOH/7-DCST/PCBM. Namely, lower hole mobility for the hole transport for PDAA/TAA/7-DCST/PCBM is significantly related to the more energetic disorder of the broader width of DOS and the scattering situation of the positional disorder. As shown in Figure 1, bulky methyl moieties attached to the phenyl group may hinder the molecular packing preferencing hole carrier hopping, and this hindrance leads to the scattering situation.

## 4. Conclusions

In this paper, we have investigated the difference in the measured photocurrent for PDAA/TPAOH/7-DCST/PCBM and PDAA/TAA/7-DCST/PCBM using a PYS measurement and analysis of the transient photocurrent using a two-trap model. The HOMO level of PDAA is evaluated in the presence of TPAOH and TAA. From the PYS measurements, the HOMO levels for PDAA/TPAOH and PDAA/TAA are evaluated to be $-5.78$ eV and $-5.79$ eV, respectively. Furthermore, the HOMO levels for PDAA/TPAOH/7-DCST/PCBM and PDAA/TAA/7-DCST/PCBM are evaluated to be $-5.80$ eV and $-5.72$ eV, respectively. It is noted that the HOMO level for PDAA/TAA/7-DCST/PCBM is considerably higher. From the analysis of the transient photocurrent based on the two-trap model, the hole mobilities for PDAA/TPAOH/7-DCST/PCBM and PDAA/TAA/7-DCST/PCBM are evaluated to be $3.3 \times 10^{-6}$ cm$^2$ V$^{-1}$ s$^{-1}$ with a $QE = 4.3 \times 10^{-3}$ and $3.9 \times 10^{-7}$ cm$^2$ V$^{-1}$ s$^{-1}$ with a $QE = 7.0 \times 10^{-4}$, respectively. The density of the shallow traps is $1.1$–$1.2 \times 10^{16}$ cm$^{-3}$ for both polymer systems and that for the deep traps is $2.0 \times 10^{16}$ cm$^{-3}$ for PDAA/TPAOH/7-DCST/PCBM. No significant difference in trap density is evaluated for either system. These values are comparable to the photorefractive number density of traps, $1.4$–$3.1 \times 10^{16}$ cm$^{-3}$, as previously reported for PDAA composites [16]. The initial photocurrent for PDAA/TPAOH/7-DCST/PCBM is simulated to be mainly governed by the transient density of the filled shallow trap, which is replaced by the transient density of the filled deep trap at a later time. However, the entire photocurrent for PDAA/TAA/7-DCST/PCBM is governed by the transient density of the shallow trap. The width of the DOS was evaluated for both polymer systems using PYS measurements. The width of the DOS for PDAA/TPAOH/7-DCST/PCBM and PDAA/TAA/7-DCST/PCBM is determined to be 0.138 eV and 0.153 eV, respectively, which represents only a small difference for both polymer composite systems. The Bässler formalism, together with the energetic and positional disorders, was used to evaluate the hole mobility for both systems. Lower hole mobility for PDAA/TAA/7-DCST/PCBM is attributed to both the energetic disorder of the broader width of DOS and the positional disorder of the scattering situation for the carrier hopping. The latter is caused by the hindrance of molecular packing due to bulky methyl moieties attached to the phenyl group.

**Author Contributions:** Conceptualization, N.T.; methodology, Y.M. and H.N.G.; software, T.S.; formal analysis, N.T.; investigation, N.T., Y.M., B.J.J., K.K., T.S., H.N.G. and W.S.; data curation, N.T.; writing—original draft preparation, N.T.; writing—review and editing, N.T. and T.S.; visualization, N.T. and T.S.; supervision, N.T.; project administration, N.T.; funding acquisition, N.T. All authors have read and agreed to the published version of the manuscript.

**Funding:** This work is supported by the Strategic Promotion of Innovative Research and Development (S-Innovation), Japan Science and Technology Agency (JST), and is partly supported by the Japan Student Services Organization (JASSO).

**Institutional Review Board Statement:** Not applicable.

**Informed Consent Statement:** Not applicable.

**Data Availability Statement:** Not applicable.

**Conflicts of Interest:** The authors declare no conflict of interest.

## Appendix A

**Table A1.** Summary of the simulation results.

| | | | | | PDAA/TPAOH/7-DCST/PCBM (35/35/30/0.6) | | | | |
|---|---|---|---|---|---|---|---|---|---|
| Type | QE/$\alpha$ (cm$^{-1}$) | $\mu$ (cm$^2$ V$^{-1}$ s$^{-1}$) | $\gamma_T$ (cm$^3$ s$^{-1}$) | $T$ (cm$^{-3}$) | $\beta_T$ (s$^{-1}$) | $\gamma_M$ (cm$^3$ s$^{-1}$) | $M$ (cm$^{-3}$) | $\beta_M$ (s$^{-1}$) | $\gamma_R$ (cm$^3$ s$^{-1}$) |
| No. 1 | $1.6 \times 10^{-2}/59$ | $9.0 \times 10^{-7}$ | $1.6 \times 10^{-13}$ | $1.2 \times 10^{16}$ | 120 | $1.5 \times 10^{-16}$ | $2 \times 10^{16}$ | 0.01 | $1.0 \times 10^{-13}$ |
| No. 2 | $6.0 \times 10^{-3}/59$ | $2.3 \times 10^{-6}$ | $1.6 \times 10^{-13}$ | $1.2 \times 10^{16}$ | 200 | $2.0 \times 10^{-16}$ | $2 \times 10^{16}$ | 0.01 | $2.0 \times 10^{-13}$ |
| No. 3 | $4.3 \times 10^{-3}/59$ | $3.3 \times 10^{-6}$ | $1.6 \times 10^{-13}$ | $1.2 \times 10^{16}$ | 200 | $2.0 \times 10^{-16}$ | $2 \times 10^{16}$ | 0.01 | $2.7 \times 10^{-13}$ |
| No. 4 | $3.8 \times 10^{-3}/59$ | $3.7 \times 10^{-6}$ | $1.6 \times 10^{-13}$ | $1.2 \times 10^{16}$ | 200 | $2.2 \times 10^{-16}$ | $2 \times 10^{16}$ | 0.01 | $2.9 \times 10^{-13}$ |
| | | | | | PDAA/TAA/7-DCST/PCBM (35/35/30/0.6) | | | | |
| Type | QE/$\alpha$ (cm$^{-1}$) | $\mu$ (cm$^2$ V$^{-1}$ s$^{-1}$) | $\gamma_T$ (cm$^3$ s$^{-1}$) | $T$ (cm$^{-3}$) | $\beta_T$ (s$^{-1}$) | $\gamma_M$ (cm$^3$ s$^{-1}$) | $M$ (cm$^{-3}$) | $\beta_M$ (s$^{-1}$) | $\gamma_R$ (cm$^3$ s$^{-1}$) |
| No. 1 | $7.0 \times 10^{-4}/45$ | $3.9 \times 10^{-7}$ | $1.7 \times 10^{-13}$ | $1.1 \times 10^{16}$ | 0.2 | $1.6 \times 10^{-19}$ | $1 \times 10^{16}$ | 0.001 | $1.4 \times 10^{-13}$ |
| No. 2 | $6.4 \times 10^{-4}/45$ | $3.7 \times 10^{-7}$ | $1.5 \times 10^{-13}$ | $1.1 \times 10^{16}$ | 0.2 | $1.0 \times 10^{-19}$ | $1 \times 10^{16}$ | 0.001 | $1.2 \times 10^{-13}$ |
| No. 3 | $4.0 \times 10^{-4}/45$ | $6.2 \times 10^{-7}$ | $1.5 \times 10^{-13}$ | $1.1 \times 10^{16}$ | 0.2 | $1.0 \times 10^{-19}$ | $1 \times 10^{16}$ | 0.001 | $1.3 \times 10^{-13}$ |

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
