# Peer review of "Photoelectron Yield Spectroscopy and Transient Photocurrent Analysis for Triphenylamine-Based Photorefractive Polymer Composites"

_photonics, doi:10.3390/photonics9120996_

Round 1

Reviewer 2 Report

Please see attached for my detailed comments. Good luck!

Reviewer 3 Report

Manuscript reports on the investigation of charge transport properties in Photorefractive Polymer Composites measuring photoconductivity and transient currents. It follows previous research of authors and shows original experimental and theoretical results. Namely, there are interesting data showing the detailed analysis of transient photocurrent. Also the excellent agreement of determined hole mobility in Table 5 combining two unlike methods is surprising. The manuscript may be published in Photonics after minor revision. Authors should improve the text according following remarks.

1. Line 52: Given dimension unit (cm^2/Vs) pertains to the mobility and not (photo)conductivity.

2. The sample thickness (50 micrometers) should be given in Section 2 even though it is defined in ref. [17]. The value is important for the understanding of the text.

3. The meaning of different experimental curves plotted by black, blue, and red points in Figs. 2 and 3 should be specified in the figure captions or in the text. 

4. The solving of the equation set (3) - (8) comprising the drift and diffusion currents (3) necessitates the definition of boundary conditions fixing the surface properties of the sample. Authors should communicate this detail.

5. What does it mean the term "eclipse time" hourly used in Section 3.4? It was not defined previously.

6. In Figs. 7 and 8, there are breaks at the experimental data next to 1 s. What is the reason of such behavior? If it was due to the switch off the illumination, authors could conveniently involve this process to their simulations proving that their model consistently explains also this feature. 

7. Lines 218-219: Values above one thousand are given inconsistently. Comma should be used either everywhere or nowhere.

Typing error: 

Figure 8 caption: DOS

Round 2

Reviewer 2 Report

The authors have spent a significant amount of efforts improving the quality of the manuscript. I found the changes and reply detailed and convinced. The manuscript would be helpful for future studies and I am happy to recommend its publications on Photonics.